# Quality Appraisal of Ambulatory Oral Cephalosporin and Fluoroquinolone Use in the 16 German Federal States from 2014–2019

**DOI:** 10.3390/antibiotics10070831

**Published:** 2021-07-08

**Authors:** Gabriele Gradl, Johanna Werning, Salka Enners, Marita Kieble, Martin Schulz

**Affiliations:** 1German Institute for Drug Use Evaluation (DAPI), 10557 Berlin, Germany; j.werning@dapi.de (J.W.); s.enners@dapi.de (S.E.); m.kieble@dapi.de (M.K.); m.schulz@fu-berlin.de (M.S.); 2Department of Medicine, ABDA–Federal Union of German Associations of Pharmacists, 10557 Berlin, Germany; 3Institute of Pharmacy, Freie Universität Berlin, 12169 Berlin, Germany

**Keywords:** antibiotic utilization, regional differences, quality appraisal, cephalosporins, fluoroquinolones, prescription rates

## Abstract

Background: Despite concerns about causing bacterial resistance and serious side effects, oral cephalosporins and fluoroquinolones are still frequently prescribed in Germany. We aimed to test a method for the detection of regional quality differences in the use of oral cephalosporins and fluoroquinolones and to apply this to the German federal states. Methods: Use of antibiotics from 2014–2019 was analyzed using dispensing data from community pharmacies claimed to the statutory health insurance (SHI) funds. Quality of regional antibiotic use in 2019 was assessed by calculating indicators based on defined daily doses per 1000 SHI-insured persons per day (DID). Oral cephalosporin and fluoroquinolone use was followed by linear regression analyses. Results: The method used was suitable to find meaningful quality differences in ambulatory oral cephalosporin and fluoroquinolone use between the German federal states. In 2019, DID varied from 1.62 in Brandenburg to 3.17 in Rhineland-Palatinate for cephalosporins and from 0.47 in Brandenburg to 0.89 in Saarland for fluoroquinolones. The city-states Hamburg, Bremen, and Berlin showed highest quality with the applied indicator set. From 2014–2019, a significant decrease in utilization of oral cephalosporins was found in all federal states. During 2017–2019, all states showed a significant decline of fluoroquinolone use.

## 1. Introduction

Overuse and misuse of antibiotics have favored the development of bacterial resistance which is a public health problem worldwide [1]. Promoting the prudent use of antibiotics will continue to be part of the European Commission’s Medicines Strategy for Europe [2]. Germany is a country with low antibiotic consumption, compared both to other nations worldwide and within the European Union [3]. Numerous nationwide and regional measures have been developed in order to face the threat of microbial resistance and Germany strives to establish informative monitoring and surveillance systems as essential tools of the overall concept [4,5,6]. Guidelines are in place for the treatment of common infectious diseases which include first and second-line antibiotics [7,8].

Of particular concern is the use of cephalosporins and fluoroquinolones. Broad-spectrum penicillins, such as amoxicillin belong to antibiotics of first choice for the treatment of respiratory tract infections (RTI), whereas cephalosporins should be avoided because of their potential of provoking resistance among Gram-negative bacteria, such as selection of extended-spectrum beta-lactamases and to increase the risk of *Clostridium difficile* infections [9,10]. In addition, the serum/tissue concentration, e.g., from orally administered cefuroxime, which is required for efficient treatment of many infections is in dispute [11].

The World Health Organization (WHO) has redefined the term ‘reserve antibiotic’ as a last resort option and the newly defined so-called ‘watch group antibiotics’, among them fluoroquinolones and second and third generation cephalosporins, as to having higher resistance potential and to be recommended as first or second choice treatments only for a limited number of indications [12]. Due to their risk of severe and potentially irreversible adverse effects, fluoroquinolones are no longer recommended for minor to moderate infections and for elderly patients [13].

It has been shown that overall use of antibiotics has decreased in Germany from 2010–2018 [14]. Variation in ambulatory antibiotic prescribing between the 16 federal states in 2003 was substantial, although the reasons for these differences are not fully understood [14,15].

Compared to other European countries and in contrast to overall use, the use of cephalosporins and fluoroquinolones was in the middle range during the study period. Compared to the countries with lowest consumption, the use of cephalosporins was 80–100 times higher in Germany than in Denmark, and the use of fluoroquinolones was 2.3–3.2 times higher than in the United Kingdom. [16,17,18]. Therefore, the use of both antibiotic groups should be monitored with particular vigilance. In all German regions, cephalosporins belong to the antibiotic groups which are most commonly prescribed, although this is not consistent with guidelines for the treatment of common infections [14,17,19]. According to Schulz et al., prescription rates of antibiotics for pharyngitis/tonsillitis, scarlet fever, pneumonia, otitis media, and urinary tract infections (UTI) and the proportion of fluoroquinolones for the treatment of common infections were higher in Germany in 2009 compared to the recommendation by the European Surveillance of Antimicrobial Consumption (ESAC) [20]. However, the use of fluoroquinolones and of cephalosporins has decreased from 2010–2018 in nearly all regions across Germany [14,19].

In the context of the increase in antimicrobial resistance, drug-specific quality indicators for ambulatory antibiotic use in Europe have been developed, derived from ESAC data [21,22]. Twelve of the 22 proposed ESAC-based quality indicators were found to have face validity and could be used to better describe antibiotic use in ambulatory care and to assess the quality of national antibiotic prescribing patterns [21,22]. Based on these indicators, total ambulatory systemic antibiotic use and use over time has been analyzed [21,22].

To the best of our knowledge, no assessment of the quality of antibiotic use has been made for the German regions–that is, the federal states. We, therefore, aimed to test the method for the detection of regional differences in the use of cephalosporins and fluoroquinolones and to apply it exemplarily to the 16 federal states of Germany. An analysis of the use of oral cephalosporins and fluoroquinolones over time complements the assessment.

## 2. Results

### 2.1. Quality Assessment of Oral Antibiotic Use in 2019

Table 1 lists the nine quality indicators for ranking the federal states with respect to the use of oral antibiotics. Germany consists of 16 partly sovereign federated states, 13 are territorial states, and three are city-states (see Table 2). The 16 states were grouped into quartiles and ranked according to their overall use of oral antibiotics in 2019 (Figure 1). The following applies to all quality indicators: Values in the first quartile have the highest quality in the evaluation.

### 2.2. Use of Cephalosporins in 2019

In the states Brandenburg (1.62 DID), Berlin (1.62 DID), Saxony (1.68 DID), Saxony-Anhalt (1.74 DID), and Thuringia (1.88 DID), the prescription densities of oral cephalosporins were lower compared to all other states. In contrast, the shares of second generation cephalosporins in the use of all oral antibiotics were lowest in Berlin (16.1%), Bremen (17.7%), Saxony-Anhalt (18.2%), Hamburg (18.5%), and North Rhine-Westphalia (18.7%). The proportion of third generation cephalosporins was lowest in Bremen (1.1%), North Rhine-Westphalia (1.3%), Saarland (1.5%), Hamburg (1.7%), and Baden-Wuerttemberg (1.7%).

### 2.3. Use of Fluoroquinolones in 2019

The share of fluoroquinolones was 6.2% of all oral antibiotic use in Germany. The prescription densities of fluoroquinolones were lowest in Brandenburg (0.47 DID), Berlin (0.51 DID), Saxony (0.51 DID), Schleswig-Holstein (0.52 DID), and Thuringia (0.55 DID). The percentages of fluoroquinolones of all oral antibiotics dispensed showed a different ranking. In Bremen (4.9%), Schleswig-Holstein (5.4%), North Rhine-Westphalia (5.5%), Hamburg (5.6%), and Berlin (5.8%) the proportion was lowest.

### 2.4. Assessment of Federal States Based on Quality Indicators

Hamburg ranked first with six out of nine indicator values in the first quartile and none in the fourth (6/0). Bremen and Berlin ranked second (5/1). Hamburg and Bremen also scored best in the quality indicators for cephalosporin use, and Schleswig-Holstein for fluoroquinolone use. Behind Schleswig-Holstein, Hamburg, Bremen and Berlin showed the same high quality in the use of fluoroquinolones. For Saxony-Anhalt, North Rhine-Westphalia and Saarland considerable differences concerning quality indicators were found: In Saxony-Anhalt, the indicator values J01 DID, J01D DID, and J01DC % were in the first quartile, but J01DD %, J01MA % and J01M SV in the fourth. Conversely, in North Rhine-Westphalia the indicator values J01DC %, J01DD % and J01MA % were in the first quartile and J01 DID, J01 SV and J01MA SV in the fourth. In Saarland, the indicator values J01DD % and J01D SV were in the first quartile and J01 DID, J01D DID, J01MA DID, and J01MA % in the fourth.

### 2.5. Changes in the Use of Cephalosporins from 2014–2019

Figure 2 shows monthly cephalosporin DID in Germany and the calculated trend component. The percentage differences between 2019 and 2014 values (range −13.7–29.2%) and average monthly changes for the 16 federal states and Germany are listed in Table 3. In all federal states, the average monthly DID of cephalosporins decreased significantly (range −0.007–0.018).

### 2.6. Changes in the Use of Fluoroquinolones from 2014–2019

Figure 3 shows the monthly DID of fluoroquinolones with the trend component running in two sections. The percentage differences between 2019 and 2014 values, average monthly changes for the periods 2014–2016 and 2017–2019 for the federal states are listed in Table 4. In the period 2014–2016, a significant decrease in average monthly fluoroquinolone DID was observed only in the federal states Baden-Wuerttemberg (−0.007 DID), Rhineland Palatinate (−0.005 DID), Hamburg (−0.004 DID), Mecklenburg West Pomerania (−0.004 DID), Thuringia (−0.003 DID), Bavaria (−0.003 DID), and Schleswig-Holstein (−0.003 DID). In the period 2017–2019, by contrast, a decline was significant in all regions, from −0.029 DID in Saarland to −0.012 DID in Brandenburg.

## 3. Discussion

For the first time, we detected both quantitative and qualitative meaningful differences between the 16 federal states in the ambulatory use of oral cephalosporins and fluoroquinolones.

### 3.1. Quality of Oral Cephalosporin Use

Our analysis shows that oral cephalosporins were less frequently dispensed in Germany in 2019 than for example in the high-use country Greece (7.3 DID), but more frequently than for example in the United Kingdom (0.02 DID), Netherlands (0.03 DID), or Denmark (0.03 DID) [18]. There were large differences between the German federal states. Due to their potential to generate antimicrobial resistance and their low bioavailability oral cephalosporins are only considered drugs of second choice for RTI including pneumonia in guidelines [8,25,26]. Nevertheless, in Germany, an increasing preference of prescribing oral cephalosporins could be observed in the years 1997–2015, especially for RTI [27]. Seasonal variation of ambulatory antibiotic use of ≥30% was considered high in a study of antibiotic use in 26 European countries and was presumably related to diagnostic labelling of bacterial RTI such as bronchitis instead of the common cold or influenza [28]. High seasonal variation in DID may thus suggest that a large proportion of oral cephalosporins were prescribed for RTI, and consequently most likely not according to guideline recommendations and thus inappropriately, in the respective states.

Our comparison of quality indicators for the use of cephalosporins between the federal states revealed that the three city-states Bremen, Hamburg, and Berlin performed better compared to the territorial states. The proportion of second and third generation cephalosporins had a strong influence on the evaluation. These substances belong to the watch group antibiotics as defined by the WHO which are only indicated for specific, limited number of infections and should be prioritized as key targets of stewardship programs and monitoring [12].

The seasonal variation in the use of cephalosporins also showed considerable differences between the federal states. Of note, the higher quality in use of oral cephalosporins in the city-states was not correlated with low use in this substance class, nor with the overall use of oral antibiotics. The observation of a higher quality of prescription of oral cephalosporins in urban regions is interesting and deserves further investigation.

According to the Advisory Council on the Assessment of Developments in the Health System, ensuring the provision of sustainable, efficient and effective ambulatory healthcare in sparsely populated, structurally disadvantaged regions is a challenge in Germany, and especially rural areas may be threatened by a lower number of general practitioners (GPs) per inhabitants, compared to cities for example [29]. An evaluation of the National Association of SHI-physicians showed that the density of all physicians (physicians per 100,000 inhabitants) was higher in the city-states (+38%) than in the territorial states in 2019 [24]. The density of specialists such as ear, nose and throat (ENT) specialists (+45%), urologists (+24%), and pediatricians (+21%) was also considerably higher in the city-states than in the territorial states [24]. One could assume that there is a correlation between physician density and the quality of antibiotic prescribing. However, the fact that the density of GPs did not differ very much between the federal states in 2019 speaks against a direct connection, as most antibiotics in 2018 were prescribed by GPs [19]. We also did not see a correlation between the density of pharmacists and quality of antibiotic prescribing in 2019. Although the two city-states of Hamburg and Berlin had a high density of pharmacists per 100,000 inhabitants (79.4 and 76.3), Saarland had the highest (84.9), where our analysis has shown that the quality of cephalosporin and fluoroquinolone prescribing was rather poor. In Brandenburg the density of pharmacists per 100,000 inhabitants was lowest (49.4), but quality of antibiotic prescribing was in the middle range.

It can be speculated that various other factors play a role, such as the patient’s expectations or time pressure during the consultation [30,31] A study on influences on the medical prescription of antibiotics in Germany has shown that indication, diagnosis and guidelines are important factors, but the doctor–patient relationship and the pharmaceutical industry also have an influence [32].

Economic considerations, on the other hand, are unlikely to have had any impact on the change in the quality of antibiotic prescribing for cephalosporins. According to the Act to Strengthen Competition in Statutory Health Insurance (GKV-WSG) from 2007, the prescribing behavior of doctors in Germany from an economic point of view is, above all, controlled by rebate contracts for generic drug products closed between manufacturers and health insurance funds [33]. In addition, cephalosporins are low-cost drugs. Their net cost per daily dose hardly changed in the years 2016–2019 (J07D: 2.0–2.1 EUR/DDD) and their overall share in the total cost of all prescribed medicinal products was marginal (0.3–0.4%) [34]. The net cost of penicillins, which are often considered as first line therapy, was in a comparable range during 2016–2019 (J07C: 1.9–2.1 EUR/DDD) [34].

Encouragingly, the trend analysis of oral cephalosporin DID during 2014–2019 showed a significant reduction in all federal states. This is in line with the observation of Holstiege et al. that the use of cephalosporins has declined in Germany during the period 2010–2018, predominantly due to a reduction in ambulatory prescriptions for pediatric patients and, potentially, the introduction of pneumococcal vaccination for children in 2006 [14,35]. In the 2019 guideline on antibiotic therapy for ENT infections, the use of cefuroxime was restricted compared to the 2008 guideline. For example, this agent is no longer recommended for use in otitis media acuta [24]. Since cefuroxime represents the most frequently dispensed oral cephalosporin in Germany, this could be another explanation for the observed decline in the dispensing of cephalosporins.

### 3.2. Quality of Oral Fluoroquinolone Use

Fluoroquinolones are contraindicated in mild to moderate infections in Germany since April 2019 due to their serious side effects [13]. The guideline on antibiotic therapy for ENT infections was adapted to this effect in 2019 [26]. Our analysis shows that Germany has moved further away from the European high-use countries such as Romania (3.1 DID) and Greece (3.0 DID) in the use of fluoroquinolones in 2019 towards a low-use country such as Norway (0.3 DID) [18]. We found that the quality of use of this drug class in 2019 was highest in Schleswig-Holstein, followed by the three city-states Hamburg, Bremen, and Berlin. An analysis by Schulz et al. showed that in 2009 there were large differences in the use of fluoroquinolones between the federal states, and that in Schleswig-Holstein the share of medical practices that fulfilled the indicator of ‘maximum quinolone prescription rate of 5%’ was highest [20]. Prescription quality for fluoroquinolones appears to have improved in the federal states during the study period (data not shown), particularly in Mecklenburg West Pomerania and Baden-Wuerttemberg. There were still substantial differences in use of oral fluoroquinolones between Schleswig-Holstein and Saarland. As above with the oral cephalosporins, it can be assumed that several factors influence quality of fluoroquinolone prescribing. Again, no correlation with the density of GPs or pharmacists could be found based on 2019 values. Rather, it stands to reason that the prescription prevalence according to the guidelines as well as the relevant prescription quality has improved at different rates in the individual federal states. It can be assumed that for fluoroquinolones prescription quality was influenced by similar factors as for cephalosporins.

In the case of fluoroquinolones as well, economic considerations are unlikely to have played a role in their prescription. Fluoroquinolones are also low-cost drugs. Net costs have not changed in the period 2016–2019 (2.6 EUR/DDD) and also account for a very small share of the total costs of all prescribed medicinal products (0.1–0.2%).

Trend analysis showed a clear change in trend. For the years 2014–2016, we found a significant decline in seven states, i.e., Baden-Wuerttemberg, Bavaria, Hamburg, Mecklenburg West Pomerania, Rhineland Palatinate, Schleswig-Holstein, and Thuringia. In the period 2017–2019, a significant downward trend was observed in all federal states. Tran observed a comparable decline in fluoroquinolone prescribing to privately insured patients in the United States after the US Food and Drug Administration (FDA) removed systemic quinolones’ indications for acute, uncomplicated UTI, acute sinusitis, and acute exacerbation of chronic obstructive pulmonary disease in May 2016 [36]. These observations may demonstrate the effectiveness of measures initiated by the risk assessment procedures for fluoroquinolones such as those by the Federal Institute for Drugs and Medical Devices (BfArM) and the European Medicines Agency (EMA) in February 2017, as the beginning of the downward trend in Germany coincided with the start of the risk assessment procedure in Europe.

Although the decline in use of oral fluoroquinolones during the evaluation period is encouraging, their SV_2019_ values as a measure of potential savings in RTI were above the 5% threshold in all federal states [22]. This indicates that there is still potential for an even more restrictive use of these antibiotics.

### 3.3. Conclusions and Recommendations towards Measures for Rational Use of Antibiotics

There are meaningful quantitative differences in the ambulatory use of oral cephalosporins and fluoroquinolones between the German federal states. Utilizing drug-specific indicators revealed quality differences that could be adequately quantified. The results may be used as a benchmark and stimulus for quality improvements in different country regions, exemplified by the German federal states.

It would be worthwhile to examine the factors that have led to the striking quality difference between the city-states and the territorial states. In their study of differences in the quality of antibiotic prescriptions in Berlin in 2016, Witte and colleagues described that the frequency of antibiotic use among children and adolescents differed greatly among different nationalities. For example, the frequency was higher among children and adolescents with Lebanese, Turkish and German nationality than among those with Asian and African nationality [37]. In 2007, however, it was already shown that education is very effective in significantly reducing the unnecessary prescriptions of antibiotics by GPs for RTI [38].

Our results suggest that the campaigns that have been conducted in Germany in recent years with the aim of promoting the prescription of antibiotics by physicians according to guidelines and informing patients about their correct use have already been successful. The campaign ‘Antibiotika gezielt einsetzen’(Targeted use of antibiotics) of the Health Authorities in Hamburg or the state wide campaign by health insurers, chambers of physicians/pharmacists, associations and the Ministry of Health ‘Rationale Antibiotikaversorgung in NRW’(Rational antibiotics supply in North Rhine-Westphalia) are examples [39,40].

We believe that such campaigns are valuable tools to raise awareness of the problem of antibiotic overuse or misuse. However, their focus is to raise awareness of the issue of reducing antibiotic use in general. Specific antibiotic groups such as cephalosporins or fluoroquinolones, where there should be reductions, have not yet been addressed. The results of our study may help target campaigns in states where the potential for these savings is highest. Our assessment scheme as well as the conclusions drawn from it can probably be transferred to other countries or regions.

### 3.4. Strengths and Limitations

The major strength of this study is that it was based on dispensing data from the majority of community pharmacies in all federal states of Germany and thus from a highly representative sample of ambulatory data, representing 88% of Germany’s population. Only data from privately insured patients were not available. The proportion of privately insured persons averaged 11% during the analysis period [41]. A further strength is that the study provides consumption data for both groups of antibacterials, cephalosporins and fluoroquinolones, at national (higher level of aggregation) and regional (lower level of aggregation) levels. The use of months as time intervals to assess data over six years is also a strength. Moreover, the month-by-month analyses, which we conducted, suggested inappropriate prescribing derived from seasonal variations.

A limitation of our quality assessment is that neither patient data including indications nor data on disease severity are available in our database. This leaves important other parameters for the quality of antibiotic prescribing unconsidered, such as the use of recommended antibiotics for specific indications. Furthermore we cannot exclude that second and third line antibiotics have been prescribed to patients previously not responding to first line antibiotics or for patients at high risk of infection-related complications [42]. The quality indicators represent only relative values and compare the values within Germany. This must be taken into account especially in the case of cephalosporins, for which use is still high compared to a neighboring country, the Netherlands, where virtually no cephalosporins are prescribed [17].

## 4. Materials and Methods

### 4.1. Study Design

We conducted a longitudinal drug utilization study in the period 2014–2019, querying the database of the German Institute for Drug Use Evaluation (DAPI) containing anonymous dispensing data from community pharmacies claimed to the SHI funds, covering 88% of Germany’s population. All claims data from a representative sample of more than 80% (until June 2019) and more than 95% (from July 2019 onwards) of the community pharmacies in all 16 federal states were available. Data were extrapolated by regional factors to 100% of the SHI-insured population [43]. Prescriptions for privately insured patients are not covered by the database. Data on the indication, treatment duration, or dosages as well as data on individual patients were not available. Prescriptions from dentists and other prescriptions that could not be assigned regionally were excluded from the analyses. The number for SHI-insured persons was obtained from the Federal Ministry of Health [23]. Physician densities were calculated from data obtained from the National Association of SHI-physicians [24].

### 4.2. Measurement of Antibiotic Use

We focused on cephalosporins and fluoroquinolones and looked at ambulatory data of orally administered drugs since the vast majority of antibiotics are prescribed in the ambulatory setting [44]. The allocation of the active ingredients was based on the official version of the German Anatomical Therapeutic Chemical (ATC) classification system with defined daily doses (DDD) published by the German Institute of Medical Documentation and Information (DIMDI) [45]. In general, the DDD is the assumed average daily maintenance dose for the main indication of a drug in adults, but based on information published by DIMDI, dosages for children have been adjusted. Individual substances were analyzed according to the ATC code level 5. Antibiotic use was estimated by defined daily doses per 1,000 SHI-insured persons per day (DID) as described previously [17].

### 4.3. Assessment of the Quality of Antibiotic Drug Use

To compare the quality of oral antibiotic use, we followed the quality indicators proposed by Coenen and Adriaenssens for the comparison of European countries [21,22]. For this purpose, seven of the twelve indicators of the inter-European comparison were selected and two additional, the share of 2nd generation cephalosporins in all antibiotics, and the seasonal variation of oral cephalosporin consumption, were added. There were no claims for 4th generation oral cephalosporins, so the consumption of 3rd generation oral cephalosporins was equivalent to that of 3rd and 4th generation cephalosporins. Consumption of oral fluoroquinolones (J01MA) was chosen instead of consumption of oral quinolones (J01M) since during the study period no oral quinolones other than fluoroquinolones were dispensed. This set of nine drug-specific quality indicators (see Table 1) was adapted to assess the use of oral cephalosporins and fluoroquinolones and used to compare the corresponding quality across federal states in 2019.

### 4.4. Trend Analysis of Oral Cephalosporin and Fluoroquinolone Use

We evaluated the time course of the use of oral cephalosporins and fluoroquinolones during the study period. Trend analysis of seasonally adjusted monthly cephalosporin and fluoroquinolone DID time series were performed. Seasonality within the 12 months was assumed and determined as a multiplicative seasonal factor. For cephalosporin DID, linear regression analyses were performed to investigate associations between time as increasing calendar month and the seasonally adjusted monthly DID (y_t_) in all federal states. For these analyses, a linear relationship was assumed between time in month and DID:y_t_ = β₀ + β₁ ∙ month_t_ ∙ ε_t_

The average monthly change estimates correspond to the estimated slope β_1_. The intercept is indicated by β₀ while ε_t_ is the error. The errors are assumed to be independent. Whether the respective trend β_1_ was significantly different from 0 could be seen from the *p*-values of the corresponding *t*-tests.

For fluoroquinolone DID, segmented regression analyses were performed to investigate associations between time as increasing calendar month before and after the risk assessment in February 2017 and the seasonally adjusted monthly DID (y_t_) in all federal states. For these analyses, the following linear relationship between time in month and DID was assumed:y_t_ = β₀ + β₁ ∙ month_t since 2014_ + β₂ ∙ month_t since 2017_ ∙ ε_t_

The average monthly change estimates before 2017 correspond to the estimated slope β_1_ while the average monthly change estimates since 2017 corresponds to the sum of estimated slopes β_1_ + β_2_. The intercept is indicated by β₀ while ε_t_ is the error. The errors are assumed to be independent. Whether the respective trends β_1_ and β_2_ were significantly different from 0 could be seen from the *p*-values of the corresponding *t*-tests.

Statistical analyses were conducted using IBM SPSS 22. Statistical tests were performed at an overall alpha level of 0.05. As each federal state was considered separately, the test alpha level for each test decision is corrected using the Bonferroni-Holm method to account for multiple testing. The coefficient of determination R^2^ was used as a measure of goodness of fit.

## Figures and Tables

**Figure 1 antibiotics-10-00831-f001:**
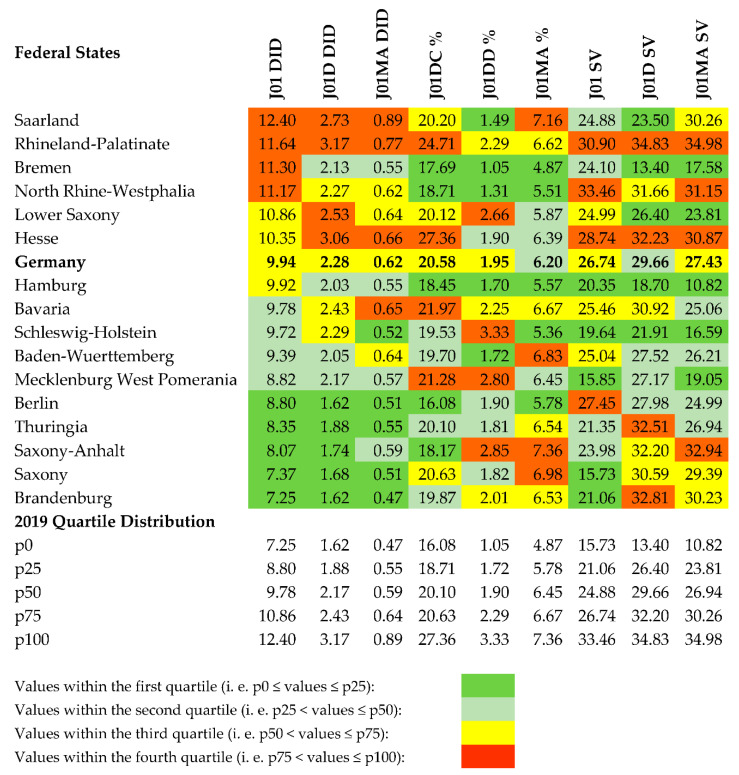
Quality indicators for ambulatory use of oral antibiotics, cephalosporins and fluoroquinolones; 2019 values for Germany and the 16 federal states grouped into quartiles based on the 2019 quartile distribution.

**Figure 2 antibiotics-10-00831-f002:**
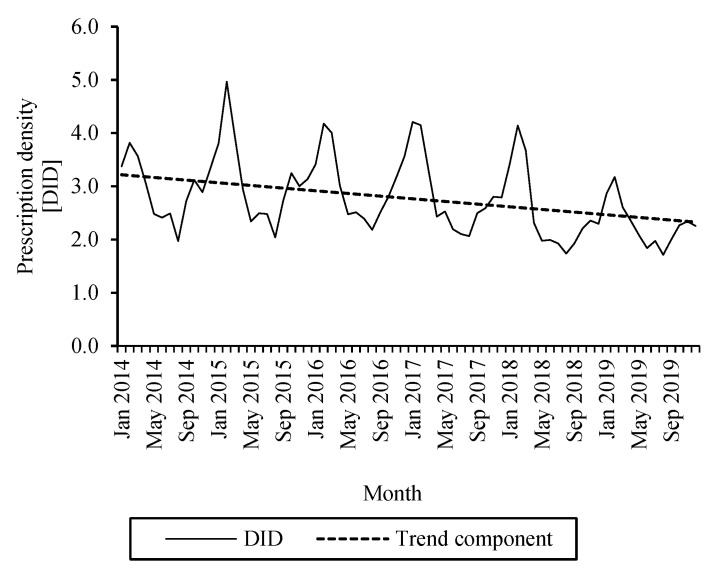
Monthly cephalosporin DID during 2014–2019 in Germany and calculated trend.

**Figure 3 antibiotics-10-00831-f003:**
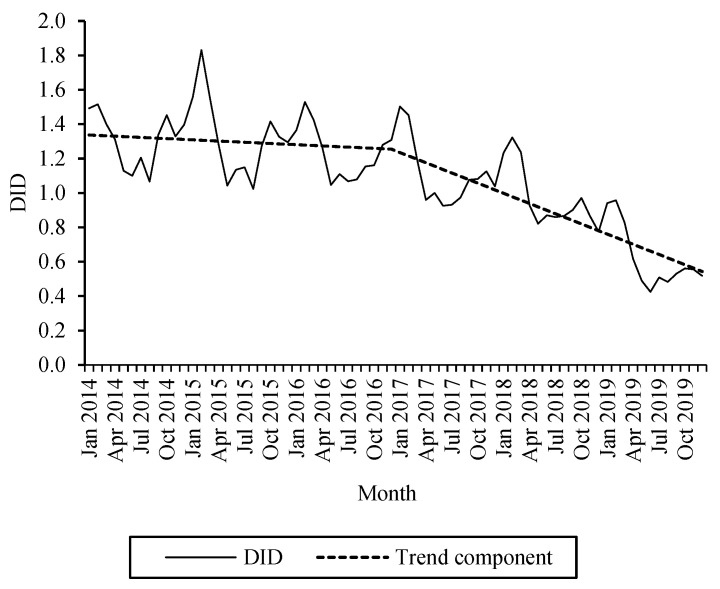
Monthly fluoroquinolone DID during 2014–2019 in Germany and calculated trend.

**Table 1 antibiotics-10-00831-t001:** Drug-specific quality indicators for oral ambulatory antibiotic use [21,22].

Label	Description
J01 DID	use of oral antibacterials for systemic use (J01), expressed in DID
J01D DID	consumption of oral cephalosporins (J01D), expressed in DID
J01MA DID	consumption of oral fluoroquinolones (J01MA), expressed in DID
J01DC %	consumption of 2nd generation oral cephalosporins (J01DC), expressed as percentage ^1^
J01DD %	consumption of 3rd generation oral cephalosporins (J01DD), expressed as percentage ^1^
J01MA %	consumption of oral fluoroquinolones (J01MA), expressed as percentage ^1^
J01 SV	seasonal variation of total oral antibiotic consumption (J01) ^2^
J01D SV	seasonal variation of oral cephalosporin consumption (J01D) ^2^
J01MA SV	seasonal variation of oral fluoroquinolone consumption (J01MA) ^2^

^1^ Percentage of total use of oral antibacterials for systemic use (J01). ^2^ Overuse in the winter quarters (October–December and January–March) compared with the summer quarters (July–September and April–June) of a 1 year period starting in July and ending in June of the following year, expressed as a percentage: [DDD (winter quarters)/DDD(summer quarters) − 1] × 100.

**Table 2 antibiotics-10-00831-t002:** Characteristics of German federal states, by state type, area, number of persons insured by statutory health insurance (SHI) funds (Source: [23]), physician density (Source: [24]) and density of community pharmacists in 2019.

Federal State	State Type TS = Territorial State CS = City-State	Area [km^2^]	Number of SHI-Insured Persons in 2019	Number of Physicians per 100,000 Inhabitants in 2019					Number of Pharmacists ^2^ per 100,000 Inhabitants in 2019
				All Physicians	General Physicians ^1^	Pediatricians	Ear-Nose- and Throat Specialists	Urologists	
Baden-Wuerttemberg	TS	35,748	9,422,699	206.8	64.9	9.9	4.8	3.7	63.8
Bavaria	TS	70,542	11,157,806	217.3	70.6	8.9	5.1	3.8	67.7
Berlin	CS	891	3,159,945	283.7	71.5	10.5	7.7	5.1	76.3
Brandenburg	TS	29,654	2,264,526	188.3	65.4	8.5	5.1	3.8	49.4
Bremen	CS	419	599,781	301.1	68.2	14.3	8.2	5.7	55.2
Hamburg	CS	755	1,597,846	292.9	72.9	11.7	7.6	4.6	79.4
Hesse	TS	21,116	5,431,318	215.2	64.6	8.4	5.1	3.7	71.7
Lower Saxony	TS	47,710	7,050,786	202.2	65.2	9.0	5.4	3.9	61.2
Mecklenburg West Pomerania	TS	23,295	1,467,844	207.1	72.8	10.5	6.5	4.3	55.6
North Rhine-Westphalia	TS	34,112	15,762,659	211.8	62.9	9.3	5.6	4.4	59.0
Rhineland-Palatinate	TS	19,858	3,442,438	195.8	66.0	9.3	4.8	4.1	70.0
Saarland	TS	2,571	855,115	221.1	66.4	9.4	6.2	4.9	84.9
Saxony	TS	18,450	3,760,346	211.6	66.4	11.2	6.3	4.6	49.8
Saxony-Anhalt	TS	20,457	2,062,627	196.4	65.1	8.9	5.8	4.6	59.4
Schleswig-Holstein	TS	15,810	2,497,277	203.6	69.9	8.9	5.1	3.8	67.9
Thuringia	TS	16,202	1,965,431	202.8	69.1	10.4	5.6	4.8	52.6

^1^ General practitioners and family physicians. ^2^ Pharmacists in community pharmacies (data provided by the Federal Chamber of Pharmacists/ABDA—Federal Union of German Associations of Pharmacists, Berlin).

**Table 3 antibiotics-10-00831-t003:** Trends for monthly cephalosporin DID during 2014–2019. Abbreviations: DID_2014_, Average DID in 2014; DID_2019_, Average DID in 2019; Δ, percentage difference between 2019 and 2014 values; Avg, average monthly change in DID. In order to account for the different alpha levels for the individual federal states due to the Bonferroni-Holm method, Avg that are significantly different from zero are marked with *; *R*^2^, coefficient of determination for the applied model.

Federal State	DID_2014_	DID_2019_	Δ	Avg	*p-*Value	*R* ^2^
Thuringia	2.65	1.88	−29.2%	−0.016	<0.001 *	0.49
Saxony	2.31	1.68	−27.3%	−0.012	<0.001 *	0.56
Baden-Wuerttemberg	2.82	2.05	−27.2%	−0.014	<0.001 *	0.74
Hesse	4.06	3.06	−24.6%	−0.018	<0.001 *	0.56
North Rhine-Westphalia	2.97	2.27	−23.4%	−0.014	<0.001 *	0.55
Berlin	2.10	1.62	−22.8%	−0.009	<0.001 *	0.58
**Germany**	**2.93**	**2.28**	**−22.1%**	**−0.013**	**<0.001 ***	**0.57**
Hamburg	2.60	2.03	−21.9%	−0.011	<0.001 *	0.56
Bremen	2.69	2.13	−20.8%	−0.010	<0.001 *	0.42
Brandenburg	2.04	1.62	−20.8%	−0.008	<0.001 *	0.49
Saxony-Anhalt	2.19	1.74	−20.5%	−0.009	<0.001 *	0.32
Bavaria	3.05	2.43	−20.3%	−0.012	<0.001 *	0.51
Mecklenburg West Pomerania	2.71	2.17	−19.8%	−0.012	<0.001 *	0.39
Rhineland Palatinate	3.83	3.17	−17.2%	−0.013	<0.001 *	0.40
Lower Saxony	3.05	2.53	−17.1%	−0.012	<0.001 *	0.35
Schleswig-Holstein	2.66	2.29	−13.9%	−0.007	<0.001 *	0.38
Saarland	3.17	2.73	−13.7%	−0.008	<0.001 *	0.22

**Table 4 antibiotics-10-00831-t004:** Trends for fluoroquinolone DID during 2014–2019. Abbreviations: DID_2014_, Average DID in 2014; DID_2019_, Average DID in 2019; Δ, percentage difference between 2019 and 2014 values; Avg, average monthly change in DID. In order to account for the different alpha levels for the individual federal states due to the Bonferroni-Holm method, Avg that are significantly different from zero are marked with *; *R*^2^, coefficient of determination for the applied model.

Federal State	DID_2014_	DID_2019_	Δ	Avg_2014–2016_	*p-*Value 2014–2016	Avg _2017–2019_	*p-*Value 2017–2019	*R* ^2^
Mecklenburg West Pomerania	1.286	0.569	−55.8%	−0.004	0.002 *	−0.016	<0.001 *	0.899
Hesse	1.488	0.661	−55.6%	−0.002	0.081	−0.021	<0.001 *	0.895
Rhineland Palatinate	1.735	0.771	−55.6%	−0.005	0.001 *	−0.021	<0.001 *	0.903
Schleswig-Holstein	1.165	0.521	−55.3%	−0.003	<0.001 *	−0.015	<0.001 *	0.940
Baden-Wuerttemberg	1.433	0.641	−55.2%	−0.007	<0.001 *	−0.012	<0.001 *	0.925
Bremen	1.201	0.550	−54.2%	0.000	0.846	−0.019	<0.001 *	0.866
Hamburg	1.203	0.553	−54.1%	−0.004	<0.001 *	−0.014	<0.001 *	0.958
North Rhine-Westphalia	1.331	0.615	−53.7%	−0.001	0.297	−0.020	<0.001 *	0.903
Thuringia	1.170	0.546	−53.4%	−0.003	0.002 *	−0.014	<0.001 *	0.901
Saarland	1.892	0.888	−53.0%	−0.002	0.237	−0.029	<0.001 *	0.885
**Germany**	**1.310**	**0.616**	**−53.0%**	**−0.002**	**0.019**	**−0.017**	**<0.001 ***	**0.918**
Berlin	1.058	0.509	−51.9%	−0.001	0.345	−0.015	<0.001 *	0.884
Saxony-Anhalt	1.213	0.594	−51.0%	0.001	0.609	−0.020	<0.001 *	0.820
Bavaria	1.329	0.652	−50.9%	−0.003	0.001 *	−0.016	<0.001 *	0.926
Saxony	1.032	0.514	−50.2%	−0.001	0.448	−0.015	<0.001 *	0.891
Brandenburg	0.929	0.474	−49.0%	−0.001	0.140	−0.012	<0.001 *	0.882
Lower Saxony	1.242	0.637	−48.7%	0.000	0.669	−0.021	<0.001 *	0.903

## Data Availability

The datasets generated during the current study are available from the corresponding author on reasonable request.

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
