# Peer review of "Quality Appraisal of Ambulatory Oral Cephalosporin and Fluoroquinolone Use in the 16 German Federal States from 2014–2019"

_antibiotics, 2021, doi:10.3390/antibiotics10070831_

Round 1

Reviewer 1 Report

The manuscript titled “Quality Appraisal of Ambulatory Oral Cephalosporin and Fluoroquinolone Use in the 16 German Federal States from 2014–2019” was generally well written and organized. The authors intended to reveal the quality differences in ambulatory oral cephalosporin and fluoroquinolone uses among the German federal states from 2014 to 2019 and a significant decrease was noticed.

There are a few comments that need to be addressed:
-​Line 46: error in quotation marks and for line 57, extra space before “Netherlands”. An additional grammar check and format proofreading for the whole article would be helpful.
-​Line 230: would recommend further discussion on the factors that influence the result
- Neither indications nor disease severity data were accessible—so adding any luck on retrieving pharmacists data (such as total number of RPh) to see if there is a correlation between pharmacist intervention and the decline in drug use?
- Also, pharmacoeconomic analysis can be considered as well to see if there is a financial change/benefit for the overall medical cost—to back up your recommendations towards measures for rational use of antibiotics.

Reviewer 2 Report

First of all, I would like to thank you for the possibility to review this paper, which highlights the very low consumption of both groups of antimicrobials (quinolones and cephalosporins) after oral administration (ambulatory use) in recent years in Germany. I would also like to make some recommendations in order to improve the presentation of the data provided in the manuscript:

  1. A comparison should be made not only with European countries which also have low DID values but also with other countries within the EU which have very high consumption values for these antimicrobials.
  2. However, the study shows that there is still a significant inter-seasonal variation in antibiotic consumption, which is related to inappropriate prescribing and should be further emphasised.
  3. The authors should provide more information on the linear relationships established for cephalosporins and quinolones. They should explain εt in the models presented, and provide the obtained values of F or the coefficient of determination or the correlation coefficient, and not only p.
  4. The study has more strengths than indicated by the authors, as it provides consumption data for both groups of antibacterials at national (higher level of aggregation) and regional (lower level of aggregation) levels. The use of monthly time periods to assess data over 6 years is also a strength of the study.
  5. On the other hand, would it be possible to venture that the campaigns implemented to reduce the use of the studied antimicrobials have worked better in the city states than in the other German federal regions?
  6. The naming of micro-organisms (Clostridium difficile) or “et al.” should be written in italics.

Reviewer 3 Report

Thank you for sharing this work. Rational use of antibiotics from 'reserve' group is crucial and this has been well presented in the manuscript. Although there are several limitations with the data presented yet, the study holds its value.

Author Response

We thank the reviewer for this very positive feedback.